# AUTOMATIC COMPLEMENTARY-SEPARATION PRUNING FOR EFFICIENT CNNS

## ABSTRACT

Reducing the complexity of neural networks without sacrificing performance is a critical challenge for deploying models in real-world, resource-constrained environments. We introduce Automatic Complementary Separation Pruning (ACSP), a novel and fully automated method for pruning convolutional neural networks that focuses on accelerating inference time. ACSP combines structured and activation-based pruning to remove redundant neurons and channels while preserving essential components. Tailored for supervised learning tasks, ACSP constructs a graph space that encodes the separation capabilities of each component across all class pairs. By leveraging complementary selection principles and clustering techniques, ACSP ensures that the selected components maintain diverse and complementary separation capabilities, reducing redundancy and maintaining high network performance. The pruning volume is determined automatically, removing the need for manual tuning. This approach significantly reduces the number of FLOPs (floating-point operations) and results in faster inference time without compromising accuracy.

## 1 INTRODUCTION

Convolutional neural networks (CNNs) have revolutionized deep learning, achieving remarkable success in applications like image classification, object detection, and image segmentation (Han et al., 2015a; Redmon, 2016; Minaee et al., 2021). However, these models, with millions of parameters, are computationally intensive, making deployment on resource-constrained devices challenging (He & Xiao, 2023). To overcome this, model compression techniques have become vital, reducing model complexity, computational demands, and memory use, and facilitating their application in real-world environments.

Among these compression techniques, pruning (Han et al., 2015b), decomposition (Denton et al., 2014), quantization (Rastegari et al., 2016), and knowledge distillation (Hinton et al., 2015) are widely studied. Pruning, in particular, removes redundant components to create more efficient, sparse networks without significant performance loss. The aim is to reduce parameters while maintaining accuracy, leading to faster inference and lower storage requirements.

Pruning techniques can be broadly categorized into two approaches: structured and unstructured pruning (Li et al., 2017). Unstructured pruning involves removing individual weights from the network based on certain criteria, such as the magnitude of the weights. While this method can lead to highly sparse models, the irregularity of the resulting network structure often limits its ability to fully leverage modern hardware acceleration (He & Xiao, 2023). This limitation arises because current hardware is optimized for dense matrix operations, meaning that the random removal of weights doesn't result in significant improvements in speed or efficiency. In contrast, structured pruning focuses on removing entire components, such as filters, channels, or layers, thereby maintaining a regular structure that is more compatible with hardware optimizations (Anwar et al., 2017). This method allows for greater speedups, as it reduces not only the parameter count but also the computational overhead in a way that is better aligned with hardware constraints (Liu et al., 2018).

Another class of pruning techniques relies on network activations, referred to as activation-based pruning. These methods prune components based on their activation during the forward pass, requiring access to the dataset on which the model was trained (Ardakani et al., 2017). By analyzing the activations, the method identifies less important components, which can be removed without

significantly degrading performance. The advantage of activation-based pruning is that it can offer more fine-grained decisions regarding which components contribute less to the network output. However, a limitation of these approaches is the need for the dataset during the pruning process.

A common challenge in many pruning techniques, regardless of the specific method, is the need for the user to manually define the size or extent of the pruning. Typically, users must specify the percentage or volume of components to be pruned, which often results in a trial-and-error process to identify the optimal pruning level. This approach not only consumes considerable time but also requires repeated evaluations to strike the right balance between model size and performance (Xiao et al., 2019). Such manual tuning diminishes the practicality of pruning in real-world applications, where time and computational resources are limited, and it hinders the scalability of pruning methods for larger networks or diverse deployment environments (Blalock et al., 2020). The need for user input in defining the pruning volume makes it difficult to achieve optimal results in an automated and efficient manner. For instance, prior works have explored automating pruning decisions – e.g., by introducing trainable gating parameters (Xiao et al., 2019) or using reinforcement learning to search pruning policies (He et al., 2018b; Liu et al., 2019). However, these methods often require complex training schemes or are limited to specific scenarios. In contrast, ACSP selects the pruning extent automatically in a single pass per layer using a data-driven knee-finding approach, without additional supervision or search.

In this paper, we introduce Automatic Complementary Separation Pruning (ACSP), a novel approach that fully automates neural network pruning. ACSP integrates both structured pruning and activation-based pruning, allowing the removal of entire components such as channels or neurons while utilizing activations to retain the most critical elements. A key concept of ACSP is its ability to select components based on their complementary capabilities, ensuring diversity and reducing redundancy in the pruned network. Unlike many conventional methods that often rely on manual user input to define the pruning volume, ACSP automatically selects the smallest and most diverse subset of components in each layer, aiming to minimize redundancy. The principle of selecting components based on complementary abilities, particularly through graph-based methods, ensures that the chosen subset contributes diverse, non-overlapping capabilities to the network. The graph-based approach avoids redundancy by selecting components from distinct regions within the graph space, ensuring that each chosen component not only performs well across tasks but also offers unique capabilities. Such complementary selection using graphs has been successfully applied in various domains, including feature selection and clustering methods (Nie et al., 2016; Zhao et al., 2022; Levin & Singer, 2024; 2025). By adopting this principle, ACSP balances efficiency with performance, enabling substantial reductions in model size without sacrificing accuracy.

The pruning process is conducted iteratively, layer by layer. For each layer, ACSP constructs a graph space based on activations, encoding the separation capability of each component with respect to all class pairs, making the method inherently suited to supervised learning tasks. To ensure complementary selection, ACSP selects components from different regions of the graph space, emphasizing diversity and complementary separation capabilities. This enables the network to maintain high performance while reducing the number of components in the neural architecture. ACSP's automated selection process uses a clustering algorithm and a knee-finding technique, making it both efficient and scalable, and therefore practical for real-world applications. In summary, the contributions of this paper can be summarized as follows:

- This paper presents ACSP, a method that automatically determines an efficient subset of components to prune without requiring manual intervention, overcoming the limitations of user-defined pruning volumes and reducing redundancy in neural networks.

- ACSP combines the strengths of structured pruning with activation-based pruning, ensuring the efficient removal of entire components like neurons or channels while selecting components with complementary separation capabilities. This approach maintains critical elements, resulting in models that are both computationally efficient and hardware-friendly.

- ACSP focuses on inference-time efficiency, removing redundant channels/neurons to yield significant speed-ups (e.g., $2.25\times$ on ResNet-50) with minimal accuracy loss.

- Extensive experiments on multiple architectures (VGG, ResNet, DenseNet, MobileNet) and datasets (CIFAR-10/100, ImageNet) show that ACSP consistently reduces computation (FLOPs) by $1.5$–$2.5\times$ while maintaining or even improving accuracy. This validates ACSP as a scalable, practical pruning solution for real-world deployment

## 2 RELATED WORK

**Structured Pruning.** Structured pruning methods focus on the removal of entire components, such as neurons, filters, or channels, creating a more streamlined and efficient network architecture that is optimized for hardware acceleration. One such method is SCOP (Scientific Control Pruning) (Tang et al., 2020), which identifies redundant structures by introducing a control group mechanism with knockoff features designed to resemble real feature maps but remain label-independent. During pruning, SCOP applies scaling factors to real and knockoff features, pruning components that rely more on knockoff features, thus minimizing the impact of irrelevant factors. Another method, SANP (Structural Alignment for Network Pruning) (Gao et al., 2023) retains alignment between the pruned and original network through partial regularization, guided by an Architecture Generator Network (AGN) that selects the optimal sub-network during training. By reducing the structural gap between the full and pruned models, SANP enhances pruning efficiency, improves hardware compatibility, and maintains high model performance. Similarly, Random Channel Pruning (Li et al., 2022b) offers a simplified approach to structured pruning by randomly selecting channels for removal. Despite its simplicity, random pruning performs comparably to more advanced techniques, particularly when paired with fine-tuning. This method effectively reduces network complexity while maintaining performance, providing a straightforward yet competitive alternative for achieving efficient neural network architectures. DepGraph (Dependency Graph) (Fang et al., 2023) introduces a dependency graph to model the dependencies between layers in neural networks, allowing for automatic group-level structured pruning. The method ensures that structurally dependent parameters across layers are pruned simultaneously, preserving network integrity. By leveraging these dependencies, Dep-Graph prunes groups of parameters, maintaining performance while reducing computational costs.

**Activation-Based Pruning.** Activation-based pruning methods rely on network activations during the forward pass to identify less important components, which are then pruned. DCP (Discrimination-aware Channel Pruning) (Zhuang et al., 2018) adds discrimination-aware losses to intermediate layers to prune channels that lack discriminative power, using activations to evaluate each channel's contribution to classification accuracy. By balancing reconstruction errors and these losses, it retains only the most valuable channels. A greedy algorithm then selects and optimizes the channels, compressing the model while preserving or enhancing performance. Another activation-based method is Network Slimming (Liu et al., 2017), which uses L1 regularization on batch normalization scaling factors, which control channel activations, to induce sparsity. Channels with small scaling factors (and thus lower activations) are pruned. After pruning, the model is fine-tuned to recover or improve accuracy. ThiNet (Luo et al., 2017) prunes entire filters from convolutional layers based on their contribution to the next layer's activations. Instead of using current-layer information, it evaluates next-layer activations to guide pruning. This pruning method reduces model size while retaining the original structure.

However, none of the above methods fully automate the choice of pruning extent – they typically require a user-defined pruning ratio or iterative sensitivity analysis. Furthermore, existing methods do not explicitly enforce diversity among kept components. These gaps motivate our proposed ACSP method, which automatically determines layer-wise pruning levels and selects complementary components via a graph-based criterion.

## 3 METHODOLOGY

### 3.1 NOTATION

Let $F(D; W)$ denote a neural network, where $D$ is the dataset and $W$ represents the weights. We consider a dataset $D = (X, Y)$, with input data $X$ and labels $Y$, where $Y$ has $C$ unique classes.

For a given network, let $L_i$ represent the $i$-th layer with weights $W_i$. The number of components (such as neurons in a linear layer or channels in a convolutional layer) in layer $L_i$ is denoted by $N_i$. The activations of layer $L_i$ are marked as $A_i$. Let $\mathcal{I}i = \{1, 2, \ldots, N_i\}$ represent the set of indices for the components in layer $L_i$, with $\mathcal{I}i, j$ denoting the $j$-th component. For a convolutional layer, the activation $A_{i,j}[t]$, the output of the $j$-th component, is an activation map of size $p \times p$, for the $t$-th sample, where $p$ represents the spatial dimensions. For a linear layer, $p = 1$, making $A_{i,j}[t]$ a scalar. The pruning process aims to find, for each layer $L_i$, a subset of the original components $\mathcal{I}_i$ that preserves the network's performance while reducing its size.

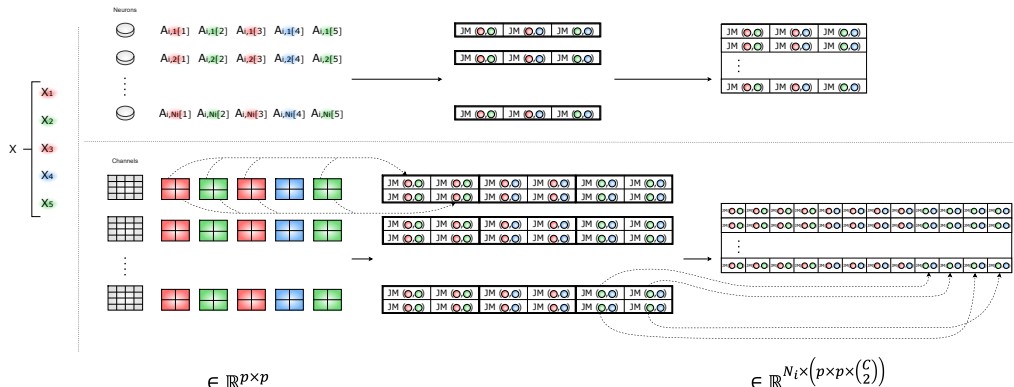

Figure 1: Building the separation matrix for a single layer, which defines the graph space: the upper part for a linear layer, and the lower for a convolutional layer. [I] A set of samples $X$, each sample $x_i$ with a label (color). [II] Perform a forward pass to obtain activations for each sample. [III] Calculate a JM value for each component and class pair, forming a separability vector. [IV] Create the matrix representation, with each row representing a component's separability, forming the graph space.

## 3.2 METHOD OVERVIEW

For each layer $L_i$, we construct a graph space that encodes the separability of each component with respect to all pairs of classes in the dataset. Each component is evaluated based on its separation score for these class pairs. This information is encoded into a vector of size $1 \times (p \times p \times \binom{C}{2})$, representing the component's position in the graph space of that layer. Further details on the construction of this graph space are provided in Section 3.3.1.

To determine the size and composition of the subset of components for layer $L_i$, we assign a score to each potential subset size in the range $[2, N_i]$. Our method employs the principle of complementary selection, which selects components with diverse and complementary separability capabilities, minimizing redundancy among components with similar separability properties. To implement this, we use a clustering algorithm that selects components from different regions of the graph space, ensuring complementary separation capabilities. The quality of each subset size is evaluated using the Mean Simplified Silhouette (MSS) index (Levin & Singer, 2024), which provides a score for each tested subset size. Section 3.3.2 provides additional details on the selection and scoring process.

After scoring each subset size, we apply a knee-finding algorithm to identify the most efficient subset size. The component subset corresponding to this size is then selected from the clustering process. The Kneedle implementation runs in $\mathcal{O}(N_i^2)$ time, but with $N_i \leq 256$ the wall-clock cost is below $0.1\,$s on an RTX 6000, so ACSP adds negligible overhead. Further details of the knee-finding process are in Section 3.4.1. At this stage, all components in the layer, apart from the identified subset, are pruned. Following this, a short fine-tuning process is conducted on a portion of the dataset to acclimate the remaining layers to the pruned layer, allowing them to adjust and optimize performance with the updated network structure. The procedure is outlined in Algorithm 1.

In the following sections, we describe in detail the steps for pruning a single layer, i.e., how to select a subset of components to retain while pruning the rest. This process is applied iteratively to each layer in the network, starting from the first hidden layer to the last, excluding both the input and output layers. By following this approach, the entire model is pruned systematically layer by layer.

## 3.3 GRAPH SPACE REPRESENTATION

### 3.3.1 CONSTRUCTING THE GRAPH SPACE

The objective of this step is to construct a graph space for a given layer $L_i$, that encodes the separability of its components $\mathcal{I}i$ across all class pairs $(c, \tilde{c})$, where $1 \leq c, \tilde{c} \leq C$. For each component $\mathcal{I}i, j$, the separability information is encoded to a vector of size $1 \times (p \times p \times \binom{C}{2})$, indicating its

position in the graph space of layer $L_i$. The process of encoding the separability vector is performed differently for linear and convolutional layers. Figure 1 shows the graph-space construction process.

**Linear Layer.**   We begin by performing a forward pass of the dataset $D$ through the network to extract activations from layer $L_i$. For each sample in $X$, we obtain $N_i$ activation values, where each activation corresponds to a scalar value for every neuron $\mathcal{I}i, j$ in layer $L_i$. To quantify the separability of a neuron $\mathcal{I}i, j$ with respect to a pair of classes $(c, \tilde{c})$, we compute the Jeffries-Matusita (JM) distance (Wang et al., 2018; Tolpekin & Stein, 2009) between the activation values $A_{i,j}$ obtained from samples labeled as class $c$ and samples labeled as class $\tilde{c}$, with respect to neuron $\mathcal{I}i, j$.

The JM distance between these two groups of activations is calculated as:

$$JM_{i,j}(c, \tilde{c}) = 2\left(1 - e^{-B_{i,j}(c,\tilde{c})}\right) \tag{1}$$

where the Bhattacharyya distance $B_{i,j}(c, \tilde{c})$ is given by:

$$B_{i,j}(c, \tilde{c}) = \frac{1}{8}\frac{(\mu_{i,j,c} - \mu_{i,j,\tilde{c}})^2}{\sigma_{i,j,c}^2 + \sigma_{i,j,\tilde{c}}^2} + \frac{1}{2}\ln\left(\frac{\sigma_{i,j,c}^2 + \sigma_{i,j,\tilde{c}}^2}{2\sigma_{i,j,c}\sigma_{i,j,\tilde{c}}}\right). \tag{2}$$

Here, $\mu_{i,j,c}$ and $\sigma_{i,j,c}^2$ denote the mean and variance of activations $A_{i,j}$ for class $c$, and likewise for $\tilde{c}$. The JM distance is the separability score of neuron $\mathcal{I}_{i,j}$ between classes $c$ and $\tilde{c}$.

The process is repeated for all neurons $\mathcal{I}i, j$ in layer $L_i$ and for all class pairs $(c, \tilde{c})$. The separability values for each component $\mathcal{I}i, j$ are encoded into a vector of size $1 \times (p \times p \times \binom{C}{2})$, where $p = 1$ for linear layers. The final matrix for layer $L_i$ has dimensions $N_i \times (p \times p \times \binom{C}{2})$, where $N_i$ is the number of neurons in the layer, and each row represents the ability of one neuron to separate between all class pairs.

**Convolutional Layer.**   In convolutional layers, each sample $t$ in $X$ produces $N_i$ activation maps from layer $L_i$, where each activation map $A_{i,j}[t]$ is a $p \times p$ matrix corresponding to a filter. To compute the separability of a channel $\mathcal{I}i, j$ between classes $(c, \tilde{c})$, we extract the activation maps $A_{i,j}$ from samples labeled as classes $c$ and $\tilde{c}$.

For each pixel in these maps, we calculate the JM distance between the pixel values from samples labeled as class $c$ and samples labeled as class $\tilde{c}$, similar to the neuron-level computation in linear layers. After calculating separability for each pixel, the resulting $p \times p$ matrix is flattened into a vector of size $1 \times (p \times p)$. This is done for all class pairs, yielding a separability vector of size $1 \times (p \times p \times \binom{C}{2})$ for each channel $\mathcal{I}i, j$. The final matrix of layer $L_i$ has size $N_i \times (p \times p \times \binom{C}{2})$, where each row represents a channel's separation ability across all class pairs.

Our method is not tied to a specific separability metric and supports various alternatives.

---

**Algorithm 1** Automatic Complementary Separation Pruning

---

**Input:** Neural Network $F(D; W)$, Dataset $D = (X, Y)$

1: **for** each layer $L_i$ in $F(D; W)$ **do**
2:     $W_i \leftarrow$ extract weights from $L_i$
3:     $N_i \leftarrow$ number of components in $L_i$
4:     $A_i \leftarrow$ extract activations from $L_i$ using $D$
5:     $graph\_space \leftarrow$ construct graph space for $L_i$
6:     $S \leftarrow \emptyset$                    ▷ MSS array
7:     **for** each $k \in \{2, \ldots, N_i\}$ **do**
8:         Apply $k$-Medoids to $graph\_space$
9:         $S[k] \leftarrow$ calculate MSS
10:     **end for**
11:     $k' \leftarrow \text{KNEEDLE}(S)$
12:     $optimal\_components \leftarrow$ top-$k'$ components by weight
13:     Prune all components of $L_i$ except $optimal\_components$
14:     Fine-tune the model on $D$
15: **end for**

In our experiments, we evaluated several metrics, including the JM, Hellinger (Rüschendorf, 1985), and Wasserstein (Beran, 1977) distances, to evaluate their effectiveness in the pruning process. While all tested metrics led to significant improvements, the JM distance consistently achieved the best balance between performance and computational efficiency. Therefore, although our approach remains flexible and adaptable to different metrics, we selected the JM distance based on its superior performance, as detailed in the experiments section.

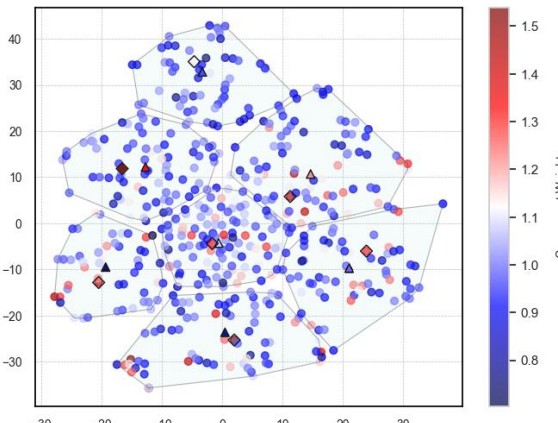

Figure 2: A 2-D view of a ResNet-56 linear layer's component space, where points denote components colored by weight. The space has 7 clusters, with medoids as triangles and highest-weight components as rhombuses. The right panel shows how per-pixel separability across class pairs forms the layer-wise separability matrix.

### 3.3.2 GRAPH SPACE EVALUATION

Our goal is to select components with complementary separation capabilities, which at the graphical level means choosing components from diverse regions of the graph space to ensure broad coverage.

Consider three components $\mathcal{I}i, j$, $\mathcal{I}i, k$, and $\mathcal{I}i, l$, each represented by their respective separation vectors in the graph space. Suppose the components $\mathcal{I}i, j$ and $\mathcal{I}i, k$ exhibit similar separation capabilities, while $\mathcal{I}i, l$ displays different separation capabilities. Graphically, this means that $\mathcal{I}i, j$ and $\mathcal{I}i, k$ are located near each other in the graph space, while $\mathcal{I}i, l$ is located in a more distant region. When selecting two components, we would prioritize selecting either $\mathcal{I}i, j$ or $\mathcal{I}i, k$ in combination with $\mathcal{I}i, l$, thereby choosing components with complementary separation capabilities. Even if $\mathcal{I}i, j$ and $\mathcal{I}i, k$ have higher separation values than $\mathcal{I}i, l$, their proximity in the graph space makes them less desirable as a pair. Instead, we aim to select one component from a different region, like $\mathcal{I}i, l$, even if its separation value is lower, as this would yield a more diverse and complementary set of separation capabilities.

To achieve this complementary selection of components, we employ the *k*-Medoids (Kaufman & Rousseeuw, 2009) algorithm. This algorithm partitions the graph space into $k$ clusters, with each cluster containing components with similar separation capabilities. After the clustering process, the medoids of the clusters are selected, representing the subset of components that provides the widest distribution of the graph space, thus satisfying the principle of complementary selection. Since the optimal value of $k$ can range from 2 to $N_i$, we must evaluate the quality of the clustering result to ensure that the selected components adequately cover the graph space. To assess this, we utilize the MSS index, which measures how well the medoids reflect this principle.

MEAN SIMPLIFIED SILHOUETTE

The Mean Simplified Silhouette (MSS) index (Levin & Singer, 2024) is used to evaluate clustering algorithms in the context of component selection, emphasizing diversity and full coverage of the graph space while minimizing redundancy. Traditional indices like the Silhouette (Rousseeuw, 1987) and Simplified Silhouette (Hruschka et al., 2006; Wang et al., 2017), assess how closely a point is associated with its cluster or medoid, and how distinct it is from the nearest non-belonging cluster. However, these methods focus only on the nearest cluster, ignoring the broader layout of the graph space, which is essential for achieving complementary selection. MSS addresses this limitation by measuring the separation between a point and all other clusters, ensuring that the selected components are not only well-associated with their cluster but also widely spread across the graph space. This ensures the chosen components complement one another and cover the space.

To compute the MSS index, we proceed as follows. For each point $i$, we define $a(i)$ as the distance between point $i$ and the center of its assigned cluster $C_h$, i.e., $a(i) = d(i, C_h)$. Next, we let $b(i)$ denote the average distance from point $i$ to the centers of all other clusters $C_l$ with $l \neq h$, so that $b(i) = \underset{l \neq h}{\text{average}}\, d(i, C_l)$. The MSS score for point $i$ is then given by $mss(i) = 1 - \frac{a(i)}{b(i)}$. Finally, the MSS index is obtained as the average of these scores across all points.

## 3.4 AUTOMATIC PRUNING

### 3.4.1 COMPONENT SIZE DEFINITION

To find a concise subset of components preserving the model's performance, we evaluate the quality of the solution for each subset size in the range $[2, N_i]$. For each size $k$ in this range, we run the $k$-Medoids algorithm on the graph space and assess the clusters using the MSS index.

After evaluating the MSS scores for all potential subset sizes, our goal is to pinpoint the point of diminishing returns, where further increases in subset size yield minimal gains in coverage. We achieve this by applying the Kneedle algorithm (Satopaa et al., 2011), which detects the "knee point" in a data curve. This knee point indicates the transition from a steep improvement to a more gradual one, signaling the most suitable size needed for optimal performance.

### 3.4.2 COMPONENT COMPOSITION DEFINITION

The Kneedle algorithm finds the knee in the MSS graph, indicating the target subset size. The cluster space for the chosen $k$ comprises $k$ medoids, reflecting the graph's broad structure and complementary selection principle. However, this selection process ignores the weights of the layer's components. Weights are critical as they signify the importance of a component to the model's performance. Higher weights indicate components that contribute to the model's predictive power. Neglecting weights in the selection process could lead to performance degradation after pruning.

To address this, we modify the selection by choosing the component with the largest weight from each cluster (for convolutional layers, we define a filter's "weight" by its $L_1$ norm, and for fully-connected layers by the absolute weight magnitude, to ensure a consistent importance metric). This ensures that we not only maintain a wide distribution of the graph space but also prioritize components with higher weights. By doing so, we preserve the model's complementary separation capabilities while retaining the components most important for performance. Figure 2 highlights the difference between these selection methods.

## 4 EXPERIMENTS

### 4.1 SETUP

We conducted experiments on CIFAR-10/100, and ImageNet-1K using VGG-16/19, DenseNet-40, MobileNet-V2, and ResNet-50/56. All models were trained to their base accuracy, then lightly fine-tuned after each layer pruning: for CIFAR-10/100, 2 epochs on a random 25% subset with learning rate 0.01 (halved after 1 epoch); for ImageNet, 3 epochs on a 25% subset with learning rate 0.003 (halved after epoch 2). This quick tune-up restores transient accuracy loss with negligible cost.

We evaluated our method using three key metrics: **Base Accuracy** (pre-pruning), **Pruned Accuracy** (post-pruning), and **Speed Up**, measured as the ratio of the number of FLOPs before and after pruning. The results of our method presented in this section were obtained using a second-degree polynomial in the Kneedle algorithm, combined with weight-based component selection.

### 4.2 CIFAR-10 RESULTS

**MobileNet-V2.** Our method yields the highest post-pruning accuracy of 94.98% with a +0.5% accuracy gain, outperforming existing approaches such as SANP (+0.45% accuracy gain) and DMC (+0.26% accuracy gain). Notably, our method also provides the best speed-up, achieving a 1.93× improvement, making it highly effective in both accuracy retention and computational efficiency.

**VGG-16.** Although AOFP achieves the highest accuracy gain (+0.46%), Our method achieves a nearly comparable accuracy improvement (+0.37%) while delivering the highest inference speed-up (2.59×). This positions our method as a well-rounded solution, balancing both accuracy and efficiency in the pruning process for VGG-16.

**ResNet-56.** Our method achieves an accuracy gain of +0.13%, which is lower than DepGraph's +0.24% improvement. However, it provides the highest speed-up at 2.15×, demonstrating its strength in computational performance. DepGraph comes close with a 2.11× speed-up but slightly surpasses us in accuracy.

| | Model | Method | Base Model | Pruned Model | Δ Accuracy | Speed Up |
|---|---|---|---|---|---|---|
| CIFAR-10 | MobileNet-V2 | DCP (Zhuang et al., 2018) | 94.47 | 94.69 | +0.22 | 1.35× |
| | | DMC (Gao et al., 2020) | 94.23 | 94.49 | +0.26 | 1.66× |
| | | SCOP (Tang et al., 2020) | 94.48 | 94.24 | -0.24 | 1.67× |
| | | ATO (Wu et al., 2024) | 94.45 | 94.78 | +0.33 | 1.84× |
| | | SANP (Gao et al., 2023) | 94.52 | 94.97 | +0.45 | 1.85× |
| | | **ACSP** | 94.48 | **94.98** | **+0.50** | **1.93×** |
| | VGG-16 | HRank (Lin et al., 2020) | 93.96 | 93.43 | -0.53 | 2.15× |
| | | GCNP (Jiang et al., 2022) | 93.10 | 93.27 | +0.17 | 2.34× |
| | | CHIP (Sui et al., 2021) | 93.96 | 93.86 | -0.10 | 2.38× |
| | | AOFP (Ding et al., 2019) | 93.38 | 93.84 | **+0.46** | 2.52× |
| | | APIB (Guo et al., 2023) | 93.68 | **94.08** | +0.40 | 2.50× |
| | | **ACSP** | 93.55 | 93.92 | +0.37 | **2.59×** |
| | ResNet-56 | CP (Li et al., 2017) | 92.80 | 91.80 | -1.00 | 2.00× |
| | | AMC (He et al., 2018b) | 92.80 | 91.90 | -0.90 | 2.00× |
| | | HRank (Lin et al., 2020) | 93.26 | 92.17 | -1.09 | 2.00× |
| | | SFP (He et al., 2018a) | 93.59 | 93.36 | -0.23 | 2.11× |
| | | DepGraph (Fang et al., 2023) | 93.53 | 93.77 | **+0.24** | 2.11× |
| | | ResRep (Ding et al., 2021) | 93.71 | 93.71 | +0.00 | 2.12× |
| | | **ACSP** | 93.69 | **93.82** | +0.13 | **2.15×** |
| CIFAR-100 | VGG-16 | DLRFC (He et al., 2022) | 73.54 | 74.09 | +0.55 | 1.76× |
| | | SCP (Kang & Han, 2020) | 73.51 | 73.86 | +0.35 | **2.06×** |
| | | **ACSP** | 73.70 | **74.31** | **+0.61** | 2.01× |
| | VGG-19 | NS (Liu et al., 2017) | 73.26 | 73.48 | +0.22 | 1.59× |
| | | SCP (Kang & Han, 2020) | 72.56 | 72.99 | +0.43 | 1.69× |
| | | SOSP (Nonnenmacher et al., 2021) | 73.45 | 73.11 | -0.34 | 2.06× |
| | | **ACSP** | 73.38 | **73.90** | **+0.62** | **2.11×** |
| | DenseNet-40 | SOSP (Nonnenmacher et al., 2021) | 74.11 | 73.46 | -0.65 | 1.42× |
| | | SCP (Kang & Han, 2020) | 74.24 | 73.17 | -1.07 | 1.86× |
| | | NS (Liu et al., 2017) | 74.64 | **74.28** | **-0.36** | 1.89× |
| | | **ACSP** | 74.30 | 73.94 | **-0.36** | **1.91×** |
| ImageNet-1K | MobileNet-V2 | CC (Li et al., 2021) | 71.88 | 70.91 | -0.97 | 1.39× |
| | | SANP (Gao et al., 2023) | 71.91 | **72.05** | **+0.14** | 1.41× |
| | | AMC (He et al., 2018b) | 71.80 | 70.80 | -1.00 | 1.43× |
| | | MetaPruning (Liu et al., 2019) | 72.00 | 71.80 | -0.80 | 1.44× |
| | | **ACSP** | 71.90 | 71.99 | +0.09 | **1.55×** |
| | ResNet-50 | HRank (Lin et al., 2020) | 76.15 | 74.98 | -1.17 | 1.77× |
| | | CHIP (Sui et al., 2021) | 76.15 | 76.30 | +0.15 | 1.81× |
| | | CCP (Peng et al., 2019) | 76.15 | **76.98** | **+0.83** | 2.04× |
| | | PaS (Li et al., 2022a) | 76.65 | 76.70 | +0.05 | 2.05× |
| | | SMCP (Humble et al., 2022) | 76.20 | 76.80 | **+0.60** | 2.15× |
| | | JMDP (Liu et al., 2021) | 76.60 | 76.00 | -0.60 | 2.15× |
| | | FPGM (He et al., 2019) | 76.15 | 75.59 | -0.56 | 2.15× |
| | | ResRep (Ding et al., 2021) | 76.15 | 76.15 | +0.00 | 2.20× |
| | | **ACSP** | 76.32 | **76.98** | +0.59 | **2.25×** |

Table 1: Pruning results on CIFAR-10/100, and ImageNet. The table reports base and pruned accuracies, accuracy change (Δ), and speed-up. Best results are in **bold**, and second-best are underlined.

## 4.3 CIFAR-100 RESULTS

**VGG-16.** Our method attains the highest post-pruning accuracy (74.31%, +0.61%), surpassing DLRFC (+0.55%) and PR (+0.42%). Although SCP provides a slightly better speed-up (2.06×), our 2.01× improvement offers a strong balance of accuracy and efficiency.

**VGG-19.** Our method again demonstrates superior performance, achieving the highest post-pruning accuracy (73.90%) with a +0.62% gain. While other methods like NS and SCP show smaller improvements, SOSP experiences a performance drop after pruning.

**DenseNet-40.** Our method provides a competitive performance with a minimal accuracy drop of −0.36%, matching NS in accuracy retention. In terms of speed-up, our method slightly edges out other approaches with a 1.91× improvement, making it the most efficient in this comparison.

## 4.4 IMAGENET-1K RESULTS

**MobileNet-V2.** Our method achieves 71.99% post-pruning accuracy (+0.09%). Though SANP achieves a slightly larger gain (+0.14%), our approach yields the highest speed-up (1.55×).

**ResNet-50.** Our method demonstrates excellent performance on ResNet-50, achieving the highest

| Dataset | Model | Batch Inference | | | Single Inference | | |
|---|---|---|---|---|---|---|---|
| | | Full Model (ms) | Pruned Model (ms) | Δ Time (%) | Full Model (ms) | Pruned Model (ms) | Δ Time (%) |
| CIFAR-10 | MobileNet-V2 | 5.339 | 4.249 | -20.39 | 3.785 | 3.686 | -2.62 |
| | VGG-16 | 1.091 | 0.975 | -10.63 | 0.771 | 0.718 | -6.88 |
| | ResNet-56 | 4.431 | 4.230 | -4.54 | 3.995 | 3.877 | -2.95 |
| CIFAR-100 | VGG-16 | 0.979 | 0.933 | -4.70 | 0.794 | 0.756 | -4.79 |
| | VGG-19 | 1.114 | 1.007 | -9.61 | 0.938 | 0.902 | -3.83 |
| | DenseNet-40 | 4.425 | 4.186 | -5.40 | 3.924 | 3.689 | -5.99 |
| ImageNet-1K | MobileNet-V2 | 7.636 | 6.814 | -10.76 | 6.203 | 5.861 | -5.51 |
| | ResNet-50 | 5.255 | 4.923 | -6.32 | 4.616 | 4.244 | -8.07 |

Table 2: Inference latency (ms) for full and pruned models under batch and single-input modes. ΔTime denotes percentage latency reduction after pruning. Results are means over 100 runs.

speed-up ($2.25\times$) among all approaches. In terms of accuracy gain, our method is second to CCP ($+0.83\%$ gain) with a $+0.66\%$ accuracy improvement. Other methods, such as CHIP and SMCP, also show competitive accuracy gains but fall short of our method's computational efficiency.

### 4.5 INFERENCE TIME ANALYSIS

Table 2 reports batch and single inference times (seconds) for full and pruned models. Values are scaled by $\times10^{-3}$ for readability, and the $\Delta$ columns show the percentage difference between them.

**Experimental Setup.** Inference times were averaged over 100 runs with random inputs, preceded by a warm-up phase to stabilize measurements. Input sizes follow dataset standards: CIFAR-10/100 use $32\times32\times3$ images, and ImageNet-1K uses $224\times224\times3$. Batch size was 40 for batch inference and 1 for single inference. Batch inference measures *throughput*, i.e., GPU efficiency on multiple inputs, while single inference measures *latency*, the time for one image to pass through the model. Experiments ran on a system with four NVIDIA Quadro RTX 6000 GPUs (24GB each).

**Results Overview.** The pruned models demonstrate consistent improvements in inference times across all datasets and architectures, reflecting the effectiveness of the ACSP pruning method. For CIFAR-10, MobileNet-V2 achieved the largest reduction in batch inference time at $-20.39\%$. Single inference times for CIFAR-10 also improved, with VGG-16 reducing latency by $-6.88\%$, while ResNet-56 showed balanced improvements for both batch ($-4.54\%$) and single ($-2.95\%$) inference.

For CIFAR-100, while the improvements were generally more modest, VGG-16 and DenseNet-40 achieved significant reductions in single inference times, with $-4.79\%$ and $-5.99\%$, respectively. VGG-19 showed the least reduction in batch inference ($-9.61\%$) but maintained consistent single inference performance at $-3.83\%$. For ImageNet-1K, the larger input size ($224\times224\times3$) inherently leads to longer inference times, as expected. However, ACSP still achieved significant reductions, with MobileNet-V2 improving batch inference by $-10.76\%$ and ResNet-50 achieving $-8.07\%$ for single inference. These results highlight ACSP's scalability to complex models and large datasets.

On average across all datasets, the pruned models demonstrated an improvement of $-8.78\%$ for batch inference and $-5.56\%$ for single inference. These reductions underscore ACSP's capability to balance computational efficiency and latency without compromising model accuracy, as demonstrated in previous sections. Notably, the wall-clock speed-ups in Table 2 are smaller than the FLOP-based factors in Table 1, as hardware utilization is not perfectly linear with FLOP count. Still, ACSP's pruned models consistently surpass full models in both throughput and latency.

## 5 CONCLUSIONS

We introduced Automatic Complementary Separation Pruning (ACSP), which automates pruning by leveraging complementary component capabilities instead of manual thresholds. Across diverse architectures, ACSP lowers computational cost, speeds up inference, and maintains or improves accuracy, making it an efficient choice for real-world deep learning applications.

A limitation of ACSP is computational overhead: building the separation graph requires comparing all class pairs, so cost scales with classes $C$ and may bottleneck for large $C$. Future work will explore approximations, such as class-pair sampling or graph-space dimensionality reduction, to reduce this dependency.

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
