## A  ANALYSIS OF OUR METHOD

The influence of polynomial degree on the pruning process is illustrated in Figure 3, which shows the impact of different polynomial degrees on the accuracy of the VGG-16 model on the CIFAR-10 dataset and the corresponding remaining Floating Point Operations Per second (FLOPs). As the polynomial degree increases, we observe a gradual reduction in the remaining FLOPs, indicating more aggressive pruning. While this is desirable in terms of computational efficiency, it comes at the cost of model accuracy, which diminishes as the polynomial degree grows. This trend demonstrates that higher-degree polynomials cause over-pruning and notable accuracy degradation. Based on this finding, ACSP fixes the knee-finding polynomial to degree 2, which (as shown by our experiments) achieves aggressive pruning with minimal accuracy loss. We found degree 2 to consistently provide the best balance between FLOP reduction and accuracy retention across architectures and datasets, so we adopt it as the default. This standardized, automatic approach eliminates the need for a trial-and-error process in determining the pruning volume, ensuring that ACSP reliably and efficiently finds the optimal subset size and composition in each layer.

Table 3 illustrates a comparison across various datasets and architectures to explain the rationale behind choosing a polynomial degree of 2 within the knee-finding algorithm. This comparison is solely provided to illustrate how different polynomial degrees influence the balance between model accuracy and pruning volume, guiding our choice of degree 2 to achieve this optimal balance. It is important to note that the knee-point polynomial degree is not a component of the method itself but rather an external decision that supports ACSP's effectiveness. The table further includes two selection methods: the "Regular" method, which selects the $k$ medoids identified at the end of clustering, and the "Weighted" method, which selects the $k$ components with the highest weights, one from each cluster.

**Method.**  The weighted method consistently outperforms the regular method across all combinations of datasets, architectures, and polynomial degrees. This highlights the significance of incorporating network weights in the component selection process.

**Polynomial degree.**  As the polynomial degree increases, the number of remaining FLOPs (%) decreases, reflecting more aggressive pruning. The polynomial degree influences the knee value the Kneedle algorithm returns: higher degrees result in smaller knee values, leading to fewer components remaining after pruning. However, higher polynomial degrees also result in greater compromises in the model's accuracy after pruning. As evidenced by Table 3, and aligning with our discussion in Section 3.4.1, a polynomial degree of 2 yields the most pruning possible without hurting – and sometimes even improving – the model's accuracy. Thus, ACSP uses degree 2 as the default setting for knee detection.

| Dataset | Model | Method | Base Accuracy | Base FLOPs | Polynum 2 Accuracy | Polynum 2 FLOPs | Polynum 3 Accuracy | Polynum 3 FLOPs | Polynum 4 Accuracy | Polynum 4 FLOPs | Polynum 5 Accuracy | Polynum 5 FLOPs |
|---|---|---|---|---|---|---|---|---|---|---|---|---|
| **CIFAR-10** | MobileNet-V2 | Regular Weighted | 94.48 | - | 94.51 94.98 | 51.81 | 90.12 90.71 | 39.62 | 87.87 88.82 | 30.80 | 86.34 87.95 | 24.07 |
| | VGG-16 | Regular Weighted | 93.55 | - | 93.55 93.92 | 38.50 | 89.44 90.01 | 29.28 | 86.98 87.83 | 22.89 | 85.33 86.97 | 17.76 |
| | ResNet-56 | Regular Weighted | 93.69 | - | 93.41 93.82 | 46.51 | 89.28 90.21 | 35.40 | 86.93 87.97 | 27.66 | 85.03 86.89 | 23.50 |
| **CIFAR-100** | VGG-16 | Regular Weighted | 73.70 | - | 73.95 74.31 | 49.75 | 70.69 71.08 | 36.14 | 68.67 69.68 | 30.38 | 67.57 68.79 | 24.10 |
| | VGG-19 | Regular Weighted | 73.38 | - | 73.36 73.90 | 47.39 | 69.95 70.89 | 38.91 | 68.30 68.83 | 33.63 | 67.16 68.00 | 27.34 |
| | DenseNet-40 | Regular Weighted | 74.30 | - | 73.51 73.94 | 52.35 | 70.23 70.85 | 39.62 | 69.13 69.34 | 30.84 | 67.91 68.75 | 23.98 |
| **ImageNet-1K** | MobileNet-V2 | Regular Weighted | 71.90 | - | 71.83 71.99 | 64.51 | 68.46 69.19 | 49.33 | 66.51 67.34 | 38.63 | 65.54 66.41 | 32.09 |
| | ResNet-50 | Regular Weighted | 76.32 | - | 76.40 76.98 | 44.44 | 73.29 73.98 | 36.77 | 71.36 72.02 | 28.51 | 69.61 71.15 | 24.56 |

Table 3: Performance comparison of accuracy and remaining FLOPs (%) across different polynomial approximation levels, using regular (medoid selection) and weighted (highest-weight selection) component selection methods.

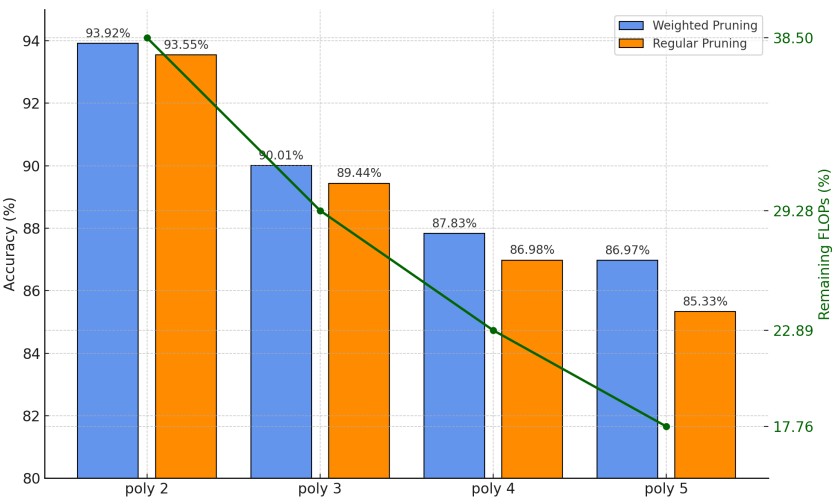

Figure 3: The effect of polynomial degree on the VGG-16 model on CIFAR-10 dataset. Higher polynomial degrees lead to fewer remaining FLOPs (%) but with a greater loss in accuracy.