# OpenReview forum: "Automatic Complementary-Separation Pruning for Efficient CNNs"
_ICLR.cc/2026/Conference — ICLR 2026 Conference Withdrawn Submission_

### Official Review · Reviewer_Dciy · 2025-10-30

**Soundness:** 2
**Presentation:** 1
**Contribution:** 2
**Rating:** 2
**Confidence:** 3

**Summary:**

The paper proposes a pruning method for neural networks based on combining structured pruning with activation-based pruning, using graph representations that encode separation capabilities of each component across class pairs.

**Strengths:**

- Using graph representations and optimizing these for pruning is a worthwhile research direction.

- The approach taken by the method seems reasonable.

**Weaknesses:**

The method described in the paper could be a useful contribution to the field, but unfortunately the current paper is lacking in several important aspects.

- The overall structure of the paper is unclear in several places. For example, in the introduction 'ACSP' is used in a sentence before the method is properly introduced and defined. Another example is 'Kneedle', which is used initially without clear introduction. In some places, experimental results are given in unexpected places (e.g. timing results in 3.2, comparison between metrics in 3.3.1).

- Several figures are unclear. Figure 1 is very difficult to read, with the caption not clearly explaining the core workings. Figure 2 lacks axis labels, and the caption mentions a right panel that does not seem to exist.

- Choices in the experimental section are not clearly motivated. How were the very specific choices in fine-tuning parameters chosen, and why do they differ between datasets and models? This makes me worry a bit about the robustness of the results -- if extensive hyperparameter tuning is needed, the method is not 'fully automatic' anymore, as the authors claim. Similarly, why are different neural network architectures used for different datasets? I.e., why aren't the same architectures used for all datasets?

- Important information in the results are missing. Importantly, what are the number of parameters of all the pruned methods? This will help assess the relative performance between compared methods. Also, how were hyperparameters chosen for the comparison methods? If these methods include user-settable hyperparameters than influence the pruning percentage, how did the authors choose these?

- In some (or many) cases, a user might want to sacrifice a bit of task accuracy to make the pruned network much faster. It is unclear whether (and if so, how) the proposed method could allow for this.

- The abstract does not include any description of experimental findings.

**Questions:**

See weaknesses above, which include several important questions.

---

### Official Review · Reviewer_7JH4 · 2025-10-31

**Soundness:** 2
**Presentation:** 2
**Contribution:** 1
**Rating:** 0
**Confidence:** 4

**Summary:**

This paper introduces Automatic Complementary-Separation Pruning (ACSP), whose core contribution is a data-driven approach to identify and preserve a diverse and essential set of network components (channels or neurons) while discarding redundant ones.

**Strengths:**

1. The writing is clear.

**Weaknesses:**

1. The Novelty is limited. The core components (structured pruning, activation-based pruning, clustering) are well-established. The main contribution is their combination.

2. The paper lacks theoretical analysis explaining why complementary selection should outperform importance-based selection. While intuitive, formal analysis of the relationship between graph space diversity and model performance would strengthen the contribution.

3. The experiments are not enough.

3.1 On ResNet-50 on Imagenet, only papers before the year 2022 are compared. The authors should include recent papers.

3.2 No runtime analysis comparing ACSP's overhead to baseline methods.

3.3 Limited Ablation Studies, like no systematic ablation on separability metrics (JM vs. Hellinger vs. Wasserstein).

4. Speedups (4-20%) are much smaller than FLOP reductions (1.5-2.5×), suggesting significant inefficiency. The authors should compare the recent paper with the realistic speedup.

5. How about the results on the vision transformer, which is recent structure?

**Questions:**

See weakness

---

### Official Review · Reviewer_KACh · 2025-11-04

**Soundness:** 2
**Presentation:** 2
**Contribution:** 2
**Rating:** 2
**Confidence:** 4

**Summary:**

The proposed Automatic Complementary Separation Pruning (ACSP), ACSP con- structs a graph space for each layer.  This space encodes the separation capabilities of each component (for e.g. channel) across all class pairs which is measure through a separation score (Jeffries-Matusita (JM) distance). Knee finding algorithm is applied to find the size of most efficient subset size. Clustering algorithm (K-mediod) is then applied on this space to find the subsets by utilizing Mean Simplified Silhouette (MSS) index. Components of the layer what are not part of any subset are finally pruned followed by fine tuning on a portion of the data. This approach reduces the number of FLOPs (floating-point operations) and results in faster inference time without compromising accuracy.

**Strengths:**

- The work uses several interesting measures and algorithms such as Jeffries-Matusita (JM) distance, Mean Simplified Silhouette (MSS), Bhattacharyya distance and knee-finding approach.

- Results clearly show that the proposed approach has achieved speed up without compromising accuracy which is encouraging. Analysis of memory footprint, flops, parameters etc would also help in improving the paper.

**Weaknesses:**

- Overall, the novelty of the work is not clear, how is the approach different from the earlier work such as Wang et al. ICIG 2019 etc. A clear note on differentiation between this work and existing structure pruning methods that are based on clustering of activation map would help in better appreciating the contributions of this work.

Wang, D., Shi, S., Bai, X., Zhang, X. (2019). Accelerating Deep Convnets via Sparse Subspace Clustering. In: Zhao, Y., Barnes, N., Chen, B., Westermann, R., Kong, X., Lin, C. (eds) Image and Graphics. ICIG 2019. Lecture Notes in Computer Science(), vol 11902. Springer, Cham. https://doi.org/10.1007/978-3-030-34110-7_50

- It is state on line 73 that “ACSP integrates both structured pruning and activation-based pruning, allowing the removal of entire components such as channels or neurons”. In my opinion structure pruning is at many times based on activations. Furthermore, while channel pruning occurs when a corresponding filter is removed and is a good example of structured pruning however typically we do not consider removal of neurons as structure pruning, it Is typically considered as unstructured pruning.

- It is stated on line 75 that “A key concept of ACSP is its ability to select components based on their complementary capabilities, ensuring diversity and reducing redundancy in the pruned network.” It is not discussed in paper how diversity aspect is addressed? How does the proposed method computes diversity?

- On 102 it is stated that the “resulting in models that are both computationally efficient and hardware-friendly.” While the computation aspect is demonstrated via speed up but hardware friendly aspect requires more investigation. Hardware friendly in case of pruning also entail efficient memory utilization, and it is generally understood that FLOPs (floating-point operations) count alone is not a reliable measure for real-world inference time, especially across different hardware architectures, analysis on this aspect may help in further improving the paper.

- Activation map of size p x p is used, is there a reason for using a square activation map.

- The paper talk about a Graph Space but from the description it seems more or less like a matrix defined for each layer that contains a separability score vector. It could be considered as a feature space or a new representation performing supspace clustering, but it is not clear, how is this a graph or graph space clustering?  A discussion on what constitute the nodes and edges, dimension of adjacency matrix, weights on edges, features on nodes etc will help in improving the understanding of the method.

- Also, please elaborate on “principle of complementary selection”, any refs?

**Questions:**

- How is the approach different from the earlier work such as Wang et al. ICIG 2019 etc. A clear note on differentiation between this work and existing structure pruning methods that are based on clustering of activation map would help in better appreciating the contributions of this work.

- How does the proposed method computes diversity?

- is there a reason for using a square activation map.

- A discussion on what constitute the nodes and edges, dimension of adjacency matrix, weights on edges, features on nodes etc will help in improving the understanding of the method.

- Analysis of memory footprint, flops, parameters etc would also help in improving the paper.

- Analysis of inference time on different hardware may also help in further improving the paper.

---

### Note · Authors · 2025-11-13

I have read and agree with the venue's withdrawal policy on behalf of myself and my co-authors.